# A CT-Based Radiomics Model for Prediction of Prognosis in Patients with Novel Coronavirus Disease (COVID-19) Pneumonia: A Preliminary Study

**DOI:** 10.3390/diagnostics13081479

**Published:** 2023-04-19

**Authors:** Lizhen Duan, Longjiang Zhang, Guangming Lu, Lili Guo, Shaofeng Duan, Changsheng Zhou

**Affiliations:** 1Department of Medical Imaging, The Affiliated Huai’an No.1 People’s Hospital of Nanjing Medical University, Huai’an 223300, China; 2Department of Medical Imaging, Jinling Hospital, School of Medicine, Nanjing University, Nanjing 210002, China; 3GE Healthcare China, Shanghai 210000, China

**Keywords:** coronavirus disease 2019, radiomics, prognosis, computed tomography

## Abstract

This study aimed to develop a computed tomography (CT)-based radiomics model to predict the outcome of COVID-19 pneumonia. In total of 44 patients with confirmed diagnosis of COVID-19 were retrospectively enrolled in this study. The radiomics model and subtracted radiomics model were developed to assess the prognosis of COVID-19 and compare differences between the aggravate and relief groups. Each radiomic signature consisted of 10 selected features and showed good performance in differentiating between the aggravate and relief groups. The sensitivity, specificity, and accuracy of the first model were 98.1%, 97.3%, and 97.6%, respectively (AUC = 0.99). The sensitivity, specificity, and accuracy of the second model were 100%, 97.3%, and 98.4%, respectively (AUC = 1.00). There was no significant difference between the models. The radiomics models revealed good performance for predicting the outcome of COVID-19 in the early stage. The CT-based radiomic signature can provide valuable information to identify potential severe COVID-19 patients and aid clinical decisions.

## 1. Introduction

Coronavirus disease 2019 (COVID-19) is a global infectious, acute respiratory illness caused by the novel severe acute respiratory syndrome coronavirus 2 (SARS-CoV-2) [1,2]. As of 1 December 2020, there have been 61.8 million confirmed cases of COVID-19 globally, including 1.4 million deaths, reported to WHO [3]. Coupled with acute respiratory syndrome (SARS) and Middle East respiratory syndrome (MERS), the three fatal coronavirus epidemics have been caused huge losses all over the world [4]. 

Real-Time quantitative polymerase chain reaction (RT-PCR) has a wide range of applications in virus detection [5]. At present, the diagnosis of COVID-19 is mainly accomplished by RT-PCR to detect SARS-CoV-2 nucleic acids [6]. Computerized tomography (CT) imaging also has high value for the diagnosis of novel coronavirus, which could manifest CT findings at different disease stages [7,8]. With CT examinations, COVID-19 patients can be sorted into four phases: early phase, progressive phase, severe phase, and dissipative phase [9,10]. The CT imaging could indicate the severity of the illness but cannot predict the development of COVID-19. Thus, finding a way to predict the development of the disease would be meaningful for clinical treatment and saving potentially critical ill patients.

Radiomics is defined as the extraction of high-throughput quantitative imaging features from images and then transforming images into mineable data that can be analyzed for decision support [11,12]. Thousands of radiomics features can be extracted from CT images. Statistical and/or machine learning methods are used to select the most valuable features to develop predictive, diagnostic, or prognostic models that may help in the diagnosis and management of various diseases. Many studies demonstrated that radiomics features can serve as effective biomarkers of COVID-19 pneumonia by reflecting the heterogeneity of lesions [13]. A retrospective cohort study by Fu et al. showed radiomic signatures of the whole lung based on support vector machines learning to evaluate the prognoses of patients with COVID-19 infection, which showed a good classification effect for the stable group and the progressive group [14]. Rizzetto et al. [15] developed an artificial intelligence tool to support radiologists in differentiating COVID-19 pneumonia from other types of viral (non-COVID-19) pneumonia using CT imaging. Hitherto, there was still no study focusing on the subtracted radiomic signatures of pulmonary lesions to predict prognosis in COVID-19. We performed this study to develop a radiomics model and the subtracted radiomics model based on chest CT in COVID-19 for predicting disease trajectories.

## 2. Material and Methods

### 2.1. Patients

A total of 44 patients (27 males and 17 females; age range, 23–74 years; mean age, 47 ± 14 years) were enrolled in this study between 24 January 2020 and 3 March 2020 at the Huai’an Fourth People’s Hospital. All patients were confirmed by RT-PCR in throat swab samples using a SARS-CoV-2 nucleic acid detection kit (Shanghai BioGerm Medical Biotechnology Co., Ltd., Shanghai, China). Conformed patients with at least 2 CT scans performed with the same model of machine were included in the analysis. The time interval between the two scans was at least 1 week but no more than 2 weeks. The exclusion criteria were as follows: (a) substantial motion artifacts in CT images; (b) negative performance in chest CT; (c) lack of follow-up; (d) coexisting or atypical CT findings suggestive of other abnormality; (e) incomplete CT datasets. Progression of disease was determined if the scope of the target lesion was enlarged or/and the solid component increased. By contrast, the disease was defined as relief if the target lesion decreased in size or/and the solid component. According to CT findings, all patients were divided into aggravate groups (*n* = 23) and relief groups (*n* = 21). All methods and the results of radiomics models are reported in accordance with the ARRIVE guidelines [16].

### 2.2. Imaging Acquisition and Segmentation

The imaging of all patients was performed with a 64-detector row CT scanner (SIEMENS SOMATOM Definition Flash-CT scanner; Siemens, Erlangen, Germany) using the following parameters: section thickness, 5 mm; acquisition field of view (FOV), 500 mm; beam pitch, 1.35 or 1.375; matrix, 512 × 512; tube voltage, 120 kV; and tube current, 60–100 mA. Each patient was provided breath-holding guidance to reduce motion-related artifacts. The images were photographed at the lung window (width, 1200 HU; level, −600 HU) and the mediastinal window (width, 350 HU (Hounsfield Units); level, 50 HU) settings. The scanning area range was from the apex to the bottom of the lung. 

CT images were obtained from the picture archiving and communication system (PACS). The slice with the largest lesion was selected. Then, the 2-dimensional region of interest (ROI) covering the center of the lesion area was segmented manually by two radiologists (with 5 and 10 years of experience, respectively) using ITK-SNAP software (version 3.8.0) (http://www.itksnap.org (accessed on 4 March 2020)). Finally, each ROI was examined by the third radiologist (with 15 years of experience).

### 2.3. Radiomics Method

The radiomics method included the following steps (Figure 1).

### 2.4. Radiomics Feature Extraction and Selection

In the first step, the image was preprocessed by resampling and gray scale discretization. The image resampling was set to 1 × 1 × 1 mm^3^, and the image gray scale was adjusted to 64 gray scales to improve the calculation speed of texture features and reduce the interference of noise. Then, AK software (AnalysisKit, GE Healthcare, Version 3.2.0) was used to extract high-throughput radiomics features from the ROIs images of all lesions in the various groups, mainly including histogram features, geometric features, texture feature, etc. All of the features were computed from the original CT images according to the “image biomarker standardization initiative” (IBSI) guidelines [17]. Since each patient had 3 ROIs, each patient had 3 sets of radiomics features. 

As for the low sensitivity of ROI segmentation, inter-observer and intra-observer agreement were examined by the third radiologist. All cases were used for feature selection and the model building. The spearman correlation test was used to calculate the correction coefficient (CC) for each feature. Features with CC > 0.8 were considered meaningful features. In order to reduce the influence of “curse of dimensionality”, we used the minimum redundancy maximum correlation (mRMR) algorithm to select the features that are most relevant to pneumonia progression [18,19]. We selected the subset of features with the mRMR method, retaining 30 features with minimum redundancy and maximum relevance. Finally, the 10-fold cross-validation of the least absolute shrinkage and selection operator (LASSO) algorithm was used to select the most useful predictive features and build the radiomic signature.

Firstly, the radiomics of the first examination between the aggravate and the relief group were compared. Then, the radiomics of the subtracted signature in the first and second examinations between the aggravate and the relief group were compared. Radiomic features of each lesion in the first and second examinations were extracted from the ROIs firstly. We subtracted the radiomics of each lesion in the two examinations. Finally, the subtracted radiomics of patients were compared between the aggravate and relief groups using the above methods. Based on the random forest (RF) algorithm, the radiomics models were built.

### 2.5. Statistical Analysis

Statistical analysis was performed using the Statistical Package for Social Sciences (SPSS) version 26.0. Clinical characteristics between different groups were analyzed using the Student’s *t*-test for continuous variables. The Kruskal-Wallis test was used to analyze the categorical variables of the two groups. The DescTools package was used to calculate ICC; the caret package was used for data grouping, Spearman correlation analysis, and calibration analysis; the glmnet package was used for LASSO regression analysis and construction of radiomics scores. Radiomics score classification was performed by the Wilcoxon test. The Mann–Whitney U test and univariate logistic regression were used for the discrimination of features between two groups. The 10-fold cross-validation was used to prove that texture analysis was valuable for discriminating one group from another group and showing the result was not due to overfitting. The pROC package was used to draw the receiver operating characteristic curve (ROC) and calculate the diagnostic performance of the model. In all tests, a *p*-value of <0.05 was considered as statistically significant. 

## 3. Results

### 3.1. Clinical Characteristics

The patients’ characteristics are summarized in Table 1. In 44 patients, the lesions ranged from 1 to 5 lobes in size and were mainly located in subpleural areas. The lesions mainly manifested as ground-glass opacities in the early stage. At an advanced stage, chest CT may demonstrate creasing consolidation in the lesions. CT imaging at the dissipation stage may show gradual resolution of the ground-glass opacity. Several cases have left fibrosis [20]. A proportion of lesions display the paving-stone sign, the air bronchi sign, and vascular thickening. There were no significant differences in clinical variables between the aggravate and relieves groups.

### 3.2. Feature Extraction and Radiomics Signature Building

In total, 782 radiomics features of the first examination were extracted from the ROIs, including 123 histogram features, 12 geometric features, and 647 texture features. Ten potential predicted radiomics features were selected by mRMR and 10-fold cross-validation of LASSO algorithm to build the first radiomics LASSO regression model for aggravate and relief groups, including three histogram features and seven texture features. A Manhattan plot was used to display the p values of those features (Figure 2). After feature selection, an AUC barplot (Figure 3A) and a heatmap (Figure 3B) were used to visualize the features selected for the radiomics model. 

A total of 782 radiomics features of the first and second examinations were extracted from the ROIs, including 140 histogram features, 12 geometric features, and 630 texture features. Ten potential predicted radiomics features were finally selected by mRMR and 10-fold cross-validation of LASSO algorithm for each patient, including six histogram features and fourtexture features. The subtracted radiomics features from the two examinations were used to build the second model. A Manhattan plot was used to display the *p* values of those features (Figure 4). After feature selection, an AUC barplot (Figure 5A) and a heatmap (Figure 5B) were used to visualize the selected features in the Radiomics model. The AUCs of the features of wavelet_LLH_firstorder_Variance and original_firstorder_Total Energy were 0.874 and 0.841, respectively. The differences between the two features are shown in Figure 6 and Figure 7.

### 3.3. Building the Prediction Model and ROC Curve Analysis

Ten suitable radiomics features were finally selected to develop the first prediction model by LASSO regression analysis. These features included wavelet_HLH_glcm_JointEnergy (*p* = 0.821), wavelet_LLL_glcm_MaximumProbability, original_glszm_ZoneEntropy (*p* = 0.812), wavelet_HLL_firstorder_Median (*p* = 0.783), wavelet_HHL_glszm_GrayLevelNonUniformity (*p* = 0.766), wavelet_HLH_glszm_GrayLevelVariance (*p* = 0.757), wavelet_LHH_glcm_JointEnergy (*p* = 0.739), wavelet_HLH_firstorder_90Percentile (*p* = 0.728), wavelet_HLH_glrlm_HighGrayLevelRunEmphasis (*p* = 0.725), and wavelet_HLH_firstorder_Maximum (*p* = 0.698). The importance values of these features are shown in Figure 8. To validate the prediction performance and reliability of the model, we also used the 100-fold leave-group-out (LGO) cross-validation technique. In addition, we instantaneously calculated and compared the sensitivity, specificity, and accuracy of the radiomics models.

Ten suitable radiomics features were selected to develop the second prediction model for predicting the prognosis of COVID-19 (*n* = 44). These features were wavelet_LLH_firstorder_Variance (*p* = 0.874), original_firstorder_TotalEnergy (*p* = 0.841), wavelet_LHH_glszm_ZonePercentage (*p* = 0.814), wavelet_LLH_firstorder_Range, wavelet_LHH_glszm_LargeAreaEmphasis (*p* = 0.790), wavelet_HLL_glszm_GrayLevelNonUniformity (*p* = 0.785), wavelet_LLL_firstorder_Maximum (*p* = 0.783), wavelet_HHH_glrlm_GrayLevelVariance (*p* = 0.780), wavelet_HLH_firstorder_Skewness (*p* = 0.773), and wavelet_HHL_firstorder_Skewness (*p* = 0.758). The importance of the selected features is shown in Figure 9. To validate the prediction performance and reliability of the models, we also used the 100-fold LGO cross-validation technique. In addition, we instantaneously calculated and compared the sensitivity, specificity, and accuracy of the radiomics models.

### 3.4. Validation of the Radiomics Signature

The performance of the first model was analyzed. The ROC is displayed in Figure 10 (AUC: 99%, 95% CI: 0.99–1.00). On the basis of the RF classifier, the accuracy, sensitivity, and specificity of the first model were 97.6%, 98.1%, and 97.2%, respectively. The accuracy, sensitivity, and specificity of the 100-fold LGO cross-validation were 83.6%, 85.0%, and 82.0%, respectively. 

The ROC of the second model is displayed in Figure 11 (AUC: 1.00, 95% CI 1.00–1.00). On the basis of the RF classifier, the accuracy, sensitivity, and specificity of the second model were 98.4%, 100%, and 97.3%, respectively. The accuracy, sensitivity, and specificity of the 100-fold LGO cross-validation were 84.5%, 81.7%, and 88.0%, respectively. 

## 4. Discussion

In this study, two machine learning-based CT radiomic signature models were constructed to effectively predict the prognosis of COVID-19 patients, which both showed good performance in the results. There was no significant difference in predictive efficiency between the two models. The first radiomics model achieved an area under the curve (AUC) of 0.99 (95% CI: 0.99–1.00), with an accuracy of 97.6%, sensitivity of 98.1%, and specificity of 97.2%. The second radiomics model achieved an AUC of 1.00 (95% CI: 1.00–1.00), with an accuracy of 98.4%, sensitivity of 100%, and specificity of 97.3%. Our findings suggest that early detection of high-risk patients at a progressive stage is feasible, allowing for the implementation of antiviral strategies or more stringent follow-up. Conversely, patients with low-risk can be safely omitted from such interventions. Overall, our study highlights the potential of machine-learning-based CT radiomic signature models for predicting the prognosis of COVID-19 patients, which may improve patient outcomes and optimize the allocation of healthcare resources. 

Radiomics is a quantitative tool for medical imaging. It provides visualized data for clinical use by extracting a large number of radiomics features and employing machine learning algorithms. Currently, this technology is widely utilized for the clinical diagnosis, staging, and prognosing of various diseases. Additionally, radiomics has shown great potential in the diagnosis, treatment, and prognosing of COVID-19 [21,22,23]. Khaniabadi et al. [24] analyzed 300 viral pneumonia patients (3-classes: 100 COVID-19, 100 pneumonia, and 100 healthy subjects) to establish a two-phase ML-based model that accurately classified COVID-19 and pneumonia patients using CT radiomics. Their results showed the model had a great potential in assessing COVID-19 CT images towards improved management of patients. Liu et al. [25] developed the automatic COVID-19 Reporting and Data System (CO-RADS) classification based on CT radiomics, which achieved an accuracy of 84% in differentiating COVID-19 and non-COVID-19 pneumonia cases.

Recent studies by Wu et al. [26] and Homayounieh et al. [27] have investigated CT-based radiomics analysis of the 3D-whole volumes of the lung to predict outcomes in COVID-19 patients, with AUCs of 0.976 and 0.99, respectively. Both studies proposed non-invasive and quantitative prognostic methods for predicting outcomes in patients. In our current study, high-resolution information was extracted from the two-dimensional CT images, enabling us to capture the intra-lesion heterogeneity and use an advanced algorithm to assess the large number of useful radiomics features. During the process of extracting radiomics, we utilized the method of subtracting the characteristics of the aggravate and relief groups to establish a new radiomics model, which yielded satisfactory results (AUC:1.00). This demonstrates that both three-dimensional and two-dimensional extraction of radiomics features have high diagnostic efficiency for predicting the prognosis of COVID-19. 

A recent study reported a combination method of the L1-norm and a support vector machine with 17 radiomics features to yield the highest performance in predicting the likelihood of rapid progression of COVID-19 pneumonia lesions on the next CT scan, with an AUC of 0.857 (95% CI: 0.766–0.947), sensitivity of 87.5%, and specificity of 70.7% [13]. In our study, the radiomics model is based on the random forest algorithm, yielding AUCs of 0.99 and 1.00, respectively. The difference in the results may be attributed to the choice of the machine learning algorithm, the size of the dataset, and the selection of specific parameters for training and evaluating the model. Both the support vector machine and the random forest are powerful machine learning algorithms and may perform well in different situations. The specific situation needs to be determined according to the dataset and task.

A recent publication by Li et al. [28] highlighted the potential of visual quantitative analysis based on CT images for accurately assessing the clinical severity of COVID-19. Consolidation was considered as one of the features of severe COVID-19 pneumonia. CT imaging at the dissipation stage may show gradual resolution of the ground-glass opacity. Several cases will leave fibrosis [20,29]. In this study, the extent of lobe involvement and that of consolidation were used as reference criteria for evaluating disease severity. Our results showed that 36 (81%) patients had the involvement of two or more lobes, which is consistent with the previous studies [30]. There were no significant differences in the range of the number of lobes involved in the aggravate and relief groups (*p* = 0.966), which can probably be attributed to the relatively small sample size (*n* = 44) in our study.

In the current study [31,32], the majority of lesions were located in the subpleural area, accounting for up to 95.5% of cases. About half of the cases presented the paving-stone sign, the air bronchi sign, vascular thickening, and fiber chords, which is consistent with previous studies [33,34,35,36]. Some literature [37,38] has reported that severe patients tend to have more consolidation, whereas non-severe patients are more likely to have more ground-glass opacities (GGO) in CT imaging. However, we did not find any significant differences in CT features between the aggravate and relief groups in our study, indicating that CT findings alone may not be a reliable predictor of the severity of COVID-19 patients [39,40].

Two radiomics models were built to assess the prognosis of COVID-19 in our study. A total of 782 radiomics features were extracted from the ROIs of each model. Ten potential predicted radiomics features were finally selected by mRMR for 10-fold cross-validation of LASSO algorithm in each model, including histogram features and texture features. These features provide intrinsic information regarding the distribution of pixel intensity and the texture morphology that is difficult for radiologists to detect [41]. For instance, wavelet features mainly reflect changes in the time and frequency domains inside the lesion. Our analysis revealed that two radiomics features, wavelet_LLH_firstorder_Variance (AUC = 0.874) and wavelet_LHH_glszm_largeAreaEmphasis (AUC = 0.841), exhibited significant differences between the aggravate and relief groups, as shown in Figure 7 and Figure 8, respectively. These findings are consistent with previous studies [42].

There are several limitations to our study. Firstly, the sample size of this study was relatively small, and thus, the conclusions might be influenced by sample bias. In order to ensure the stability of the model’s prediction performance, we used the 100-fold cross-validation method to obtain the prediction performance of the model through different training sets and took the average value of the results of 100 cycles as the prediction accuracy with this dataset. Therefore, the method is repeatable under all datasets. Secondly, clinical and laboratory variables were not included in the prediction models due to the lack of matching with each chest CT examination, which could have affected the accuracy of our predictions. Finally, the lack of uniformity in the selection of ROI for COVID-19 radiomics research limits the application of radiomics. In addition, radiomics features are known to be affected by various factors, such as the machine platform, a reconstruction algorithm, the scan sequence, and imaging parameters. Therefore, the use of machine learning in CT imaging and disease staging needs a larger sample size to increase stability and precision in the future. Moreover, huge validation on an independent testing dataset is necessary.

## 5. Conclusions

In this study, the radiomics model and subtracted radiomics model based on CT imaging have good performance in predicting the progress of COVID-19. It can provide a reference for clinical treatment. At the same time, it will provide new ideas for the research, diagnosis, and treatment of other coronavirus pneumonia.

## Figures and Tables

**Figure 1 diagnostics-13-01479-f001:**
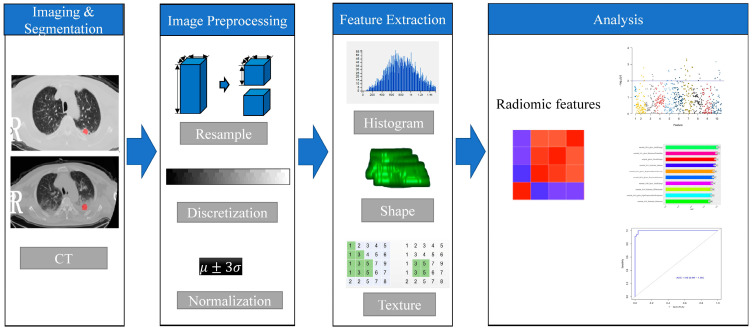
Workflow of the radiomics method. An experienced radiologist segmented the region of interest (ROI) of the lesions. Features were selected to build models. The receiver operating characteristic (ROC) was used to demonstrate diagnostic efficiency of the models.

**Figure 2 diagnostics-13-01479-f002:**
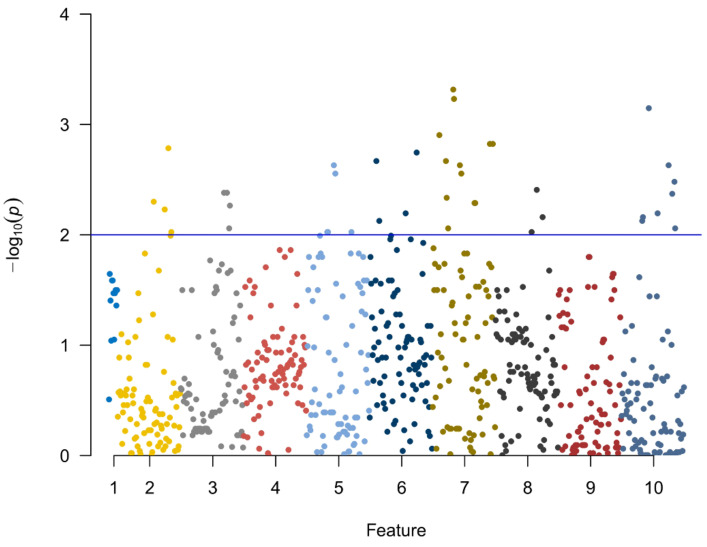
The *p* values of the 10 feature parameters and corresponding coefficients after screening in the first model are shown in this Manhattan plot.

**Figure 3 diagnostics-13-01479-f003:**
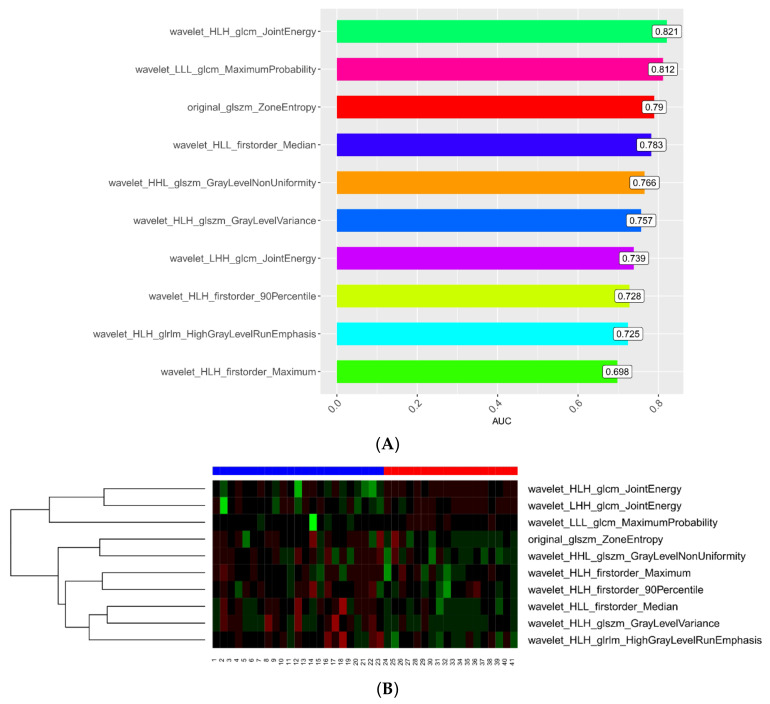
(**A**) AUC barplot showing AUCs of the features used in the first model. (**B**) The 10 feature parameters and corresponding coefficients after screening in the first model are shown in a heatmap.

**Figure 4 diagnostics-13-01479-f004:**
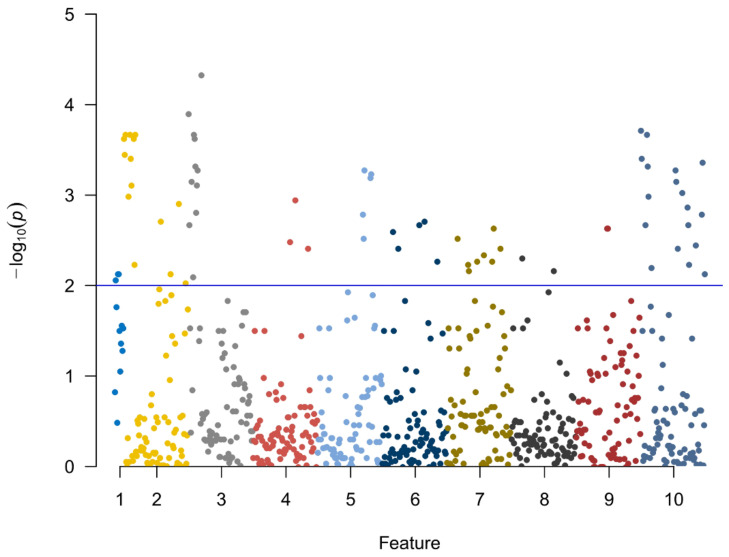
The *p* values of the 10 feature parameters and corresponding coefficients after screening in the second model are shown in this Manhattan plot.

**Figure 5 diagnostics-13-01479-f005:**
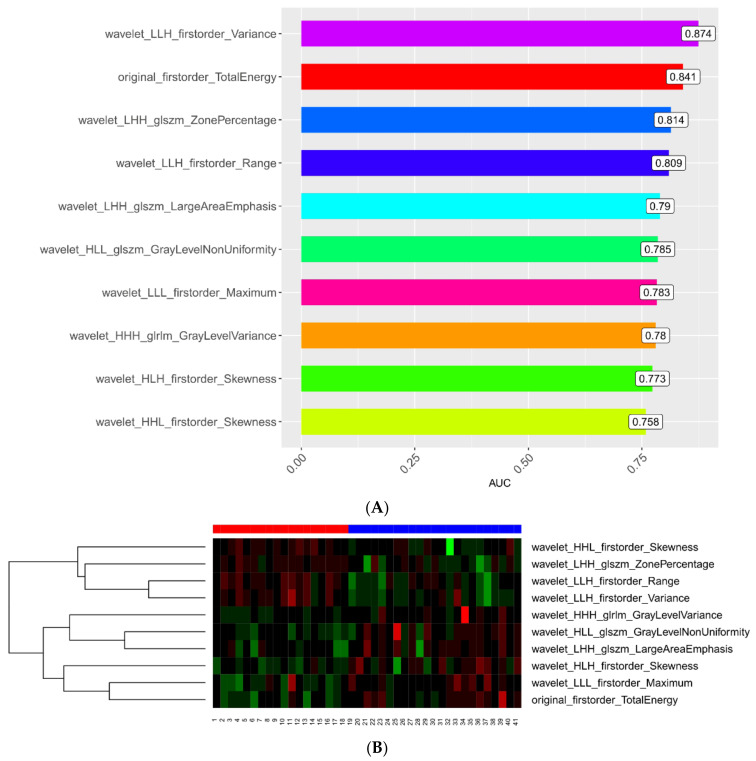
(**A**) AUC barplot showing AUCs of features used in the second model. (**B**) The 10 feature parameters and corresponding coefficients after screening in the second model in a heatmap.

**Figure 6 diagnostics-13-01479-f006:**
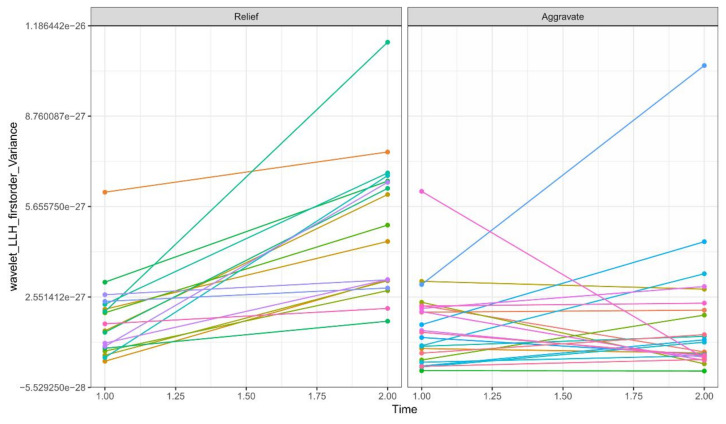
The differences in wavelet_LLH_firstorder_Variance between relief and aggravate groups.

**Figure 7 diagnostics-13-01479-f007:**
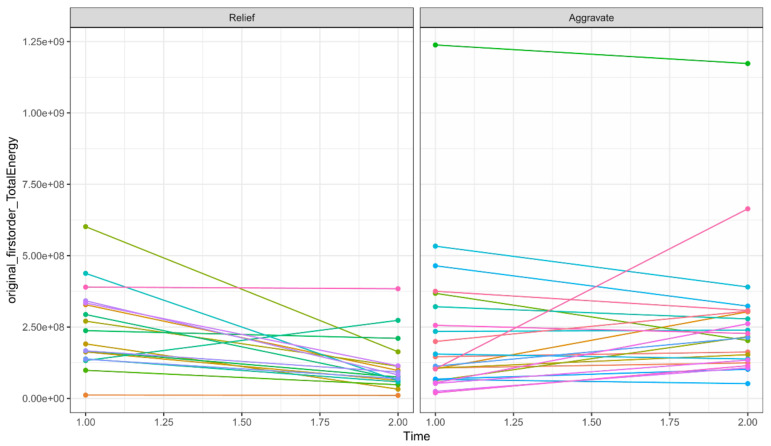
The differences of original_firstorder_TotalEnergy between relief and aggravate groups.

**Figure 8 diagnostics-13-01479-f008:**
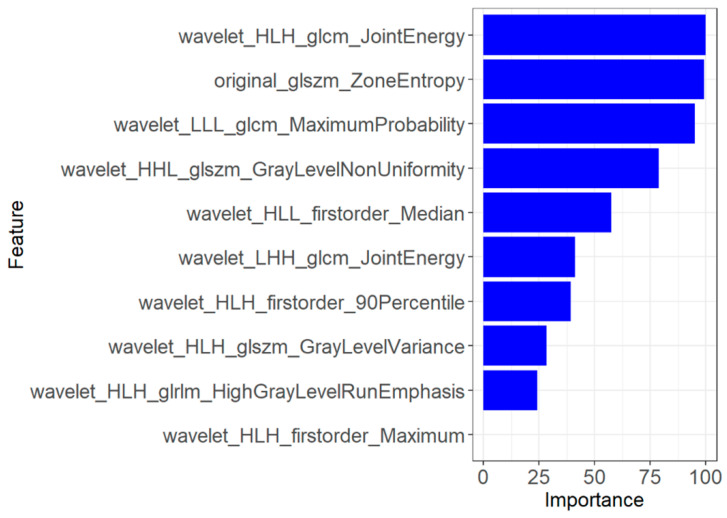
The 10 feature parameters and corresponding coefficients after screening in the first model.

**Figure 9 diagnostics-13-01479-f009:**
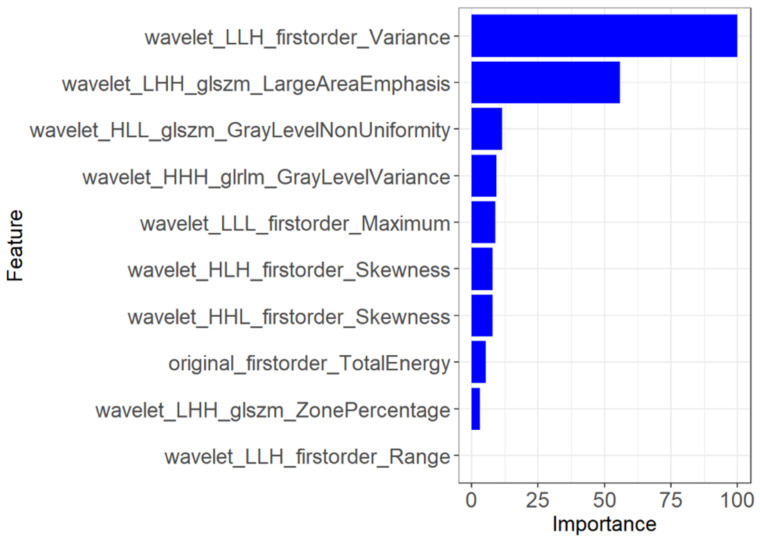
The 10 feature parameters and corresponding coefficients after screening in the second model.

**Figure 10 diagnostics-13-01479-f010:**
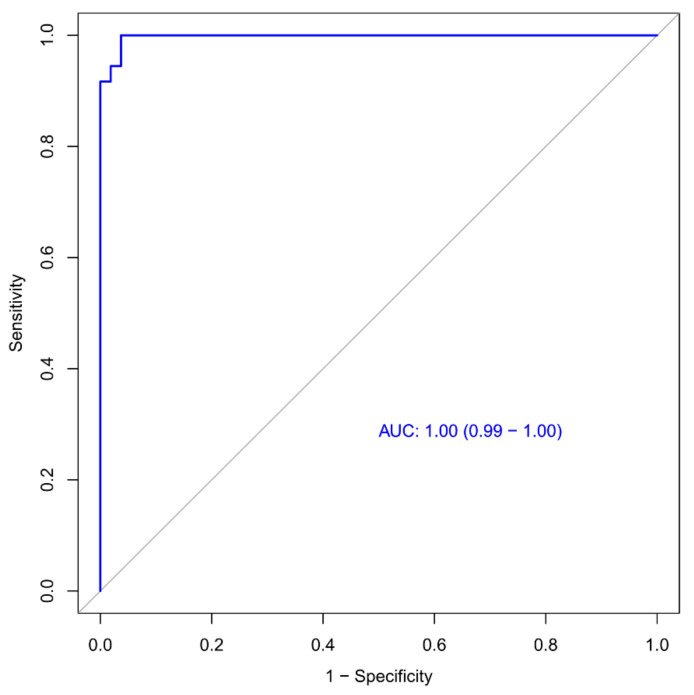
Receiver operating characteristic (ROC) curve for the first model. The AUC (area under the ROC) was 0.99 for the model.

**Figure 11 diagnostics-13-01479-f011:**
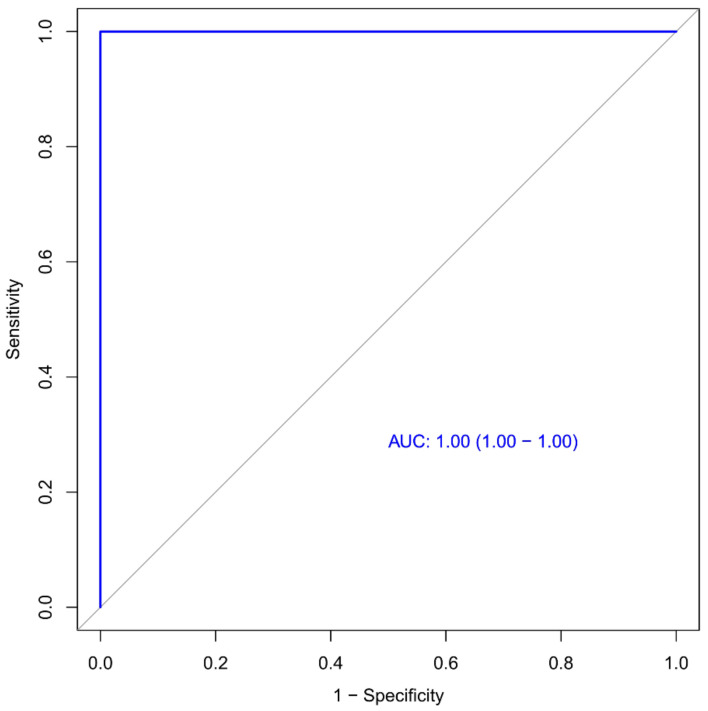
Receiver operating characteristic (ROC) curve for the second model. The AUC (area under the ROC) was 1.00 for the model.

**Table 1 diagnostics-13-01479-t001:** Clinical characteristics of patients in aggravate and relief groups.

Characteristics	Aggravate Group (*n* = 23)	Relief Group (*n* = 21)	All (*n* = 44)	*p*
Age, mean ± SD (y) *	45 ± 15	49 ± 13	47 ± 14	0.448 ^†^
Gender (male/female)	12/11	15/6	27/44	0.190 ^†^
Location of lesion, *n* (%)				0.947 ^†^
Subpleural	22 (95.7)	20 (95.2)	42 (95.5)	
others	1 (4.3)	1 (4.8)	2 (4.5)	
Range (lobes of lung), *n* (%)				0.966 ^#^
1	4 (17.4)	4 (19.0)	8 (18.2)	
2	4 (17.4)	2 (9.5)	6 (13.6)	
3	2 (8.7)	2 (9.5)	4 (9.1)	
4	5 (21.7)	5 (23.8)	10 (22.7)	
5	8 (34.8)	8 (38.1)	16 (36.4)	
Density, *n* (%)				0.123 ^#^
Ground-glass opacity	11 (47.8)	4 (19.0)	14 (31.8)	
Mix ground glass	10 (43.5)	15 (71.4)	25 (56.8)	
solid	2 (8.7)	2 (9.5)	4 (9.1)	
Paving stone sign	9 (39.1)	12 (57.1)	21 (47.7)	0.583 ^†^
Air bronchi sign	14 (60.9)	13 (61.9)	27 (61.4)	0.131 ^†^
Vascular thickening	15 (65.2)	9 (42.9)	24 (54.5)	0.232 ^†^
Fiber chords	11 (47.8)	13 (61.9)	24 (54.5)	0.349 ^†^

* Data are means ± standard deviations; ^†^ independent sample *t* test; ^#^ Kruskal-Wallis test.

## Data Availability

The data used to support the findings of this study are available from the corresponding author upon request.

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
