# Peer review of "A CT-Based Radiomics Model for Prediction of Prognosis in Patients with Novel Coronavirus Disease (COVID-19) Pneumonia: A Preliminary Study"

_diagnostics, 2023, doi:10.3390/diagnostics13081479_

Round 1
Reviewer 1 Report
Review Report
The paper entitled “A CT-based radiomics model for prediction of prognosis in patients with novel coronavirus disease (COVID-19) pneumonia” is interesting and well written but I have some suggestions to authors for its improvement as mentioned below:
1. How the method used by the authors are more accurate and reliable as compared with the previous works done, the authors should show the comparison with previous work.
2. As the authors have mentioned that the research was carried between January 2020 to March 2020, so why three years old data is used, why not tested in year 2022.
3. Some latest reference should be used as to show the latest study in this scenario.
4. Now mostly people are vaccinated, so how effective is the method to detect the disease after vaccination.
Reviewer 2 Report
Review for diagnostics-2275149
In this article entitled “A CT-based radiomics model for prediction of prognosis in patients with novel coronavirus disease(COVID-19) pneumonia ”, Duan et al. reported a developed model of CT-based radiomics that can be used in the prognosis of COVID-19 in 44 patients.
Hereafter some comments revealed after reviewing the manuscript.
1- The major lack of the study is the very low number of patients. In fact, a total of 44 only is not definitely determining such conclusion. Otherwise, it sounds primordially to add “preliminary study” even in the title itself.
2- Correct the full name of the SARS-CoV-2, which is given in line 29.
3- In line 38, add “patients” after COVID-19.
4- For the first and specifically the third sentences of the introduction, the authors may insert the following reference: Acute respiratory distress syndrome: a life threatening associated complication of SARS-CoV-2 infection inducing COVID-19. J. Biomol. Struct. Dyn. 2021, 39 (17), 6842-6851.
5- In the section “2.1. Patients”, i) as mentioned in the 1st comment, the number of the studied population is very low: 44 only. ii) the authors should specify in which specialized hospital the 44 patients were enrolled (line 61).
6- I guess the authors should insert “- CT scanner” just after “Flash” in line 75, to give the correct and the full name of the CT scanner.
7- It would be better to rewrite the conclusion and, specifically, associate it to the major findings of the study.
8- English language is fine just small editing is required, notably the punctuation in the main text but also in the subheadings. Also use the past form “performed” in line 54,
Minor comments:
9- Insert spaces before the references’ brackets in the whole manuscript. Add a space after “27” in line 59.
10- Delete the letter “s” in “materials” ; line 57.
Reviewer 3 Report
In general, the paper focus on the development of the radiomics model and the subtracted radiomics model based on chest CT in COVID-19 for predicting disease trajectories. The radiomics models revealed good performance in predicting the outcome of COVID-19 in the early stage. The CT-based radiomic signature can provide valuable information to identify potential severe COVID-19 patients and aid clinical decisions. The paper could be accepted with some suggestions for improvement as follows:
Abstract:
The limitation and research gap in the current prediction method for the outcome of COVID-19 pneumonia should be highlighted in the abstract
Introduction:
Details literature review supported with current and related references should be provided for the following points:
· Application of real-time quantitative polymerase chain reaction (RT-PCR) to detect various types of the nucleic acid compounds
· Diagnose different types of viruses by computerized tomography (CT) imaging
· Different types of abiotic features extracted from CT images
Results and discussion:
Concise and details technical discussion supported with current and related references should be provided for the following points:
· Why does the CT imaging at the dissipation stage show gradual resolution of the ground glass opacity
· How to confirm the sensitivity, specificity, and accuracy of the radionics models.
· What is the relation between the patient's clinical variables between the aggravate and relieved groups
· How to validate the prediction performance and reliability of the models
· What factor contributes on the accuracy, sensitivity, and specificity of the 100-fold LGO cross-validation
Round 2
Reviewer 1 Report
The paper is revised nicely and can be accepted.
Reviewer 2 Report
Second round report for diagnostics-2275149
Following consideration of the reviewers' comments, it can obviously be noticed that the manuscript has been improved to warrant publication in DIAGNOSTICS.
I have no further comments.